# Confocal Laser Endomicroscopy: Enhancing Intraoperative Decision Making in Neurosurgery

**DOI:** 10.3390/diagnostics15040499

**Published:** 2025-02-19

**Authors:** Francesco Carbone, Nicola Pio Fochi, Giuseppe Di Perna, Arthur Wagner, Jürgen Schlegel, Elena Ranieri, Uwe Spetzger, Daniele Armocida, Fabio Cofano, Diego Garbossa, Augusto Leone, Antonio Colamaria

**Affiliations:** 1Department of Neurosurgery, Karlsruher Neurozentrum, Städtisches Klinikum Karlsruhe, 76133 Karlsruhe, Germany; francesco.carbone615@gmail.com (F.C.); uwe.spetzger@klinikum-karlsruhe.com (U.S.); augustoleone96@gmail.com (A.L.); 2Division of Neurosurgery, Policlinico “Riuniti”, University of Foggia, 71122 Foggia, Italy; colamariaa@gmail.com; 3Department of Neurosurgery, Università degli Studi di Torino, 10125 Torino, Italy; fochinicola98@gmail.com (N.P.F.); fabio.cofano@unito.it (F.C.); diego.garbossa@unito.it (D.G.); 4Department of Neurosurgery, Klinikum rechts der Isar, Technical University Munich School of Medicine, 81675 Munich, Germany; arthur.wagner@tum.de; 5Department of Neuropathology, Klinikum rechts der Isar, Technical University Munich School of Medicine, 81675 Munich, Germany; schlegel@tum.de; 6Unit of Clinical Pathology, Advanced Research Center on Kidney Aging (A.R.K.A.), Department of Medical and Surgical Sciences, University of Foggia, 71122 Foggia, Italy; elena.ranieri@unifg.it; 7IRCCS Istituto Neurologico Mediterraneo Neuromed, 86077 Roma, Italy; danielearmocida@yahoo.it; 8Department of Neurosurgery, AOU Città della Salute e della Scienza, 10126 Torino, Italy; 9Faculty of Human Medicine, Charité Universitätsmedizin Berlin, 10117 Berlin, Germany

**Keywords:** brain tumor, confocal laser endomicroscopy, intraoperative pathology, image-guided surgery, fluorescence-guided surgery

## Abstract

Brain tumors, both primary and metastatic, represent a significant global health burden due to their high incidence, mortality, and the severe neurological deficits they frequently cause. Gliomas, especially high-grade gliomas (HGGs), rank among the most aggressive and lethal neoplasms, with only modest gains in long-term survival despite extensive molecular research and established standard therapies. In neurosurgical practice, maximizing the extent of safe resection is a principal strategy for improving clinical outcomes. Yet, the infiltrative nature of gliomas often complicates the accurate delineation of tumor margins. Confocal laser endomicroscopy (CLE), originally introduced in gastroenterology, has recently gained prominence in neuro-oncology by enabling real-time, high-resolution cellular imaging during surgery. This technique allows for intraoperative tumor characterization and reduces dependence on time-consuming frozen-section analyses. Recent technological advances, including device miniaturization and second-generation CLE systems, have substantially improved image quality and diagnostic utility. Furthermore, integration with deep learning algorithms and telepathology platforms fosters automated image interpretation and remote expert consultations, thereby accelerating surgical decision making and enhancing diagnostic consistency. Future work should address remaining challenges, such as mitigating motion artifacts, refining training protocols, and broadening the range of applicable fluorescent probes, to solidify CLE’s role as a critical intraoperative adjunct in neurosurgical oncology.

## 1. Introduction

Primary brain tumors represent an unsolved global health challenge. Their annual incidence ranges between 7 and 14.8 cases per 100,000 individuals, with a notable variation influenced by factors such as geography, socioeconomic development, sex, and age. Industrially developed countries report higher diagnosis rates, likely due to an enhanced access to healthcare and advanced imaging technologies. Tumor prevalence also demonstrates a cumulative male predominance, with a male-to-female ratio of 1.5:1, and its incidence peaks between the ages of 55 and 75 years [1,2,3]. Among all brain tumors, gliomas constitute 86% of primary brain neoplasms, with high-grade gliomas (HGGs) accounting for 30.5% of these cases. In contrast, brain metastases are significantly more prevalent than primary brain tumors, occurring in 20–40% of patients with solid tumors, making them approximately 10 times more common than primary brain malignancies [4,5,6,7].

The consequences of brain tumors are devastating not only in terms of mortality but also in the profound neurological impairments they cause, resulting in significant socioeconomic and emotional burdens on patients, families, and caregivers [8]. In global health metrics, brain tumors—primary and metastatic—rank as the 10th most burdensome malignancy in terms of disability-adjusted life years (DALYs), trailing behind other aggressive cancers such as cervical and pancreatic cancers [9]. The severe neurological deficits often associated with brain tumors—ranging from motor dysfunction and cognitive decline to personality changes—exacerbate the burden, making effective management a priority for healthcare providers and policymakers alike. Despite extensive research into the molecular mechanisms driving brain tumor development and progression, patient outcomes for aggressive brain tumors, particularly HGGs, remain poor. HGGs, classified as World Health Organization grades III and IV, are among the most lethal brain tumors. The median overall survival (OS) for grade III gliomas ranges from 24 to 72 months, while for grade IV gliomas, such as glioblastomas (GBMs), it is approximately 14 to 16 months [10]. The five-year survival rate for GBM patients is notably low, between 5% and 10% [2].

Translating molecular findings into effective therapies has been challenging. Standard treatment regimens, including maximal surgical resection, radiotherapy, and chemotherapy, have not substantially improved long-term survival rates [11,12]. From a surgical standpoint, maximal safe tumor resection is a cornerstone in the management of brain tumors. A greater yet safe extent of resection (EOR) and lower residual tumor volumes have been consistently associated with improved patient outcomes [13,14,15]. However, the general infiltrative nature of gliomas often precludes complete surgical excision, whereas metastases tend to be more easily resectable [16]. The neurosurgeon’s primary goal is to safely achieve maximal resection of malignant tissue while minimizing the risk of permanent neurological deficits, which could impede the patient’s ability to undergo essential adjuvant therapies such as radiotherapy and chemotherapy. In this context, advancements in neural and vascular tissue visualization are vital to neuro-oncological surgery [17]. These innovations enhance the surgeon’s operative capabilities and knowledge by providing detailed quantitative preoperative and intraoperative data. This improved understanding of microanatomy and functional interconnections forms the foundation of the discipline, ultimately supporting better surgical precision and patient outcomes.

## 2. Technical Advancements in Operative Visualization

Technological advancements have significantly reshaped the surgical management of brain tumors, enabling safer and more precise interventions. These innovations have revolutionized surgical techniques and improved patient outcomes by enhancing both the direct and indirect visualization of brain and tumor structures (Table 1).

One of the most transformative developments was the introduction of the surgical microscope, which marked a milestone in microsurgical techniques [18,19]. By improving the visualization of the brain–tumor interface and microanatomical structures, the surgical microscope has reduced the risk of residual tumor tissue and iatrogenic neurological deficits—key factors influencing patient prognosis [13,20]. The neurosurgical use of the microscope began in 1957 when Theodor Kurze adopted it to remove cranial nerve tumors [21]. Zeiss’s OPMI series (Zeiss, Oberkochen, Germany), introduced in the 1950s, and subsequent advancements such as motorized focus and stereoscopic systems further enhanced its functionality. Modern microscopes integrate high-definition imaging, real-time angiography, and tumor visualization technologies [22]. One of these tumor visualization technologies is represented by the use of fluorescent dyes to enhance the indirect visualization of malignant tissue and blood flow. 5-aminolevulinic acid (5-ALA), a prodrug metabolized into the fluorescent compound protoporphyrin IX, accumulates in glioma cells and fluoresces (emission wavelength: 620 to 704 nm) under blue-violet light (~405 nm), aiding in tumor identification. Fluorescein sodium (FNa), a water-soluble dye with an excitation peak around 490 nm, accumulates in regions of blood–brain barrier disruption and emits green fluorescence (~515 nm), making it effective for tumor and vascular delineation. Indocyanine green (ICG), a near-infrared dye with an excitation peak of 805 nm, binds to plasma proteins and is widely used for real-time angiography to visualize blood vessels during aneurysm clipping and bypass surgeries. Integrated into modern microscopes, these dyes provide high-contrast, dynamic imaging, facilitating the differentiation of pathological and healthy tissue, reducing surgical risks, and enhancing outcomes in both tumor resection and vascular neurosurgery [23,24,25,26]. 

More recent technological advancements have paved the way for the intraoperative direct and indirect visualization of malignant brain tissue by significantly increasing the magnification of the targeted surgical field without losing focus or image quality. The following is a comprehensive and critical review of the role of confocal laser endomicroscopy (CLE) as a novel visualization tool in the neurosurgeon’s armamentarium for the surgical management of brain tumors.

## 3. Confocal Laser Endomicroscopy: Initial Experiences

CLE has emerged as a promising intraoperative visualization tool across various medical specialties. In gastroenterology, CLE has proven effective in detecting premalignant conditions, such as Barrett’s esophagus, and malignant lesions in the upper gastrointestinal (GI) tract [27,28]. Nowadays, the Miami classification system has been adapted for esophageal CLE-based diagnosis, aiding in the identification of high-grade dysplasia and early esophageal cancer [29]. Similarly, CLE has been employed to detect early gastric cancer and premalignant conditions like gastritis and intestinal metaplasia. Additionally, CLE has demonstrated utility in identifying Helicobacter pylori infections by visualizing morphological changes in tissues [30]. In the lower GI tract, CLE facilitates the identification of adenomas and neoplastic lesions in colorectal polyps, supporting enhanced diagnostic accuracy during colonoscopy [31]. Further experiences have documented the benefits of CLE in inflammatory bowel disease and pancreatic, biliary, respiratory, and urinary systems [32,33,34,35]. 

In neurosurgery, CLE has proven to be a promising tool, providing in vivo and ex vivo, real-time histological insights during brain tumor resection (Figure 1) [36,37,38].

This technique enables high-resolution, cellular-level imaging directly in the surgical field, enhancing tumor margin detection and improving the EOR [39,40]. Brain tumor resection presents unique challenges, primarily due to the infiltrative nature of many tumors and the necessity of preserving functional brain tissue. Conventional intraoperative histological techniques, such as frozen sections, are limited by sampling errors, artifacts, and prolonged processing times, whereas CLE enables detailed imaging of cellular and subcellular structures in real time [41,42]. 

In 2011, Martirosyan et al. [43] employed CLE in murine glioma models to assess its ability to delineate tumor margins. The use of ICG in near-infrared CLE enabled the visualization of individual tumor cells and microvascular architecture. For instance, the authors were able to visualize clusters of pleomorphic, highly mitotic extravascular structures, morphologically consistent with malignant cells. Belykh et al. [44] investigated the interrater discrepancies between one neuropathologist and three neurosurgeons in classifying CLE images of murine gliomas or damaged brain tissue following the administration of FNa. Notably, the mean accuracy, specificity, and sensitivity were 90%, 86%, and 96%, respectively, with a good interobserver agreement overall. These findings underscore CLE’s diagnostic reliability, comparable to, and sometimes exceeding that of frozen section analysis. Foersch et al. [45] further validated CLE’s efficacy in differentiating various brain tumor types, such as GBMs and meningiomas, through distinct cellular morphologies, expanding the fields of application for CLE in neurosurgery (Table 2).

## 4. Advancements in CLE Technology

Recent advancements in CLE technology have significantly enhanced its clinical applications and diagnostic accuracy. These improvements focus on device miniaturization, enhanced image quality, novel scanning techniques, and integration with complementary imaging modalities. The evolution of CLE systems has been driven by the transition from first-generation (Gen1—Optiscan 5.1, Carl Zeiss AG) to second-generation (Gen2—Convivo, Carl Zeiss AG) devices, as outlined by Belykh et al. [51] Gen2 systems offer a higher image resolution, improved user interfaces, and enhanced metadata handling. However, Gen2 has a smaller field of view, which provides more pathology-like images, although it may risk hindering the surgical applicability of the findings, as a region of 475 × 267 μm is considerably limited when it comes to resective surgery. Furthermore, the screening of a wide surgical field in smaller frameworks may significantly slow the surgery itself. However, the inclusion of Z-stack imaging facilitates three-dimensional visualization of deep tissue structures, enabling more precise differentiation between normal and pathological tissues. This novel feature allows for the capture of multiple sequential image slices at different focal depths within a tissue. These slices are then compiled to create a three-dimensional visualization of the tissue structure. In addition, combining this approach with optimized fluorescent contrast agents and deep learning algorithms for real-time image interpretation further enhances the detection of infiltrative tumor margins. These integrated strategies may improve intraoperative decision making and ensure a more complete and safe tumor resection. Moreover, new-generation CLE probes have been miniaturized, allowing for greater flexibility and usability in confined surgical spaces, as highlighted by Hong et al. [52]. In their study, the authors used a miniaturized scanning CLE system (cCeLL ex vivo; VPIX Medical) for the ICG-based histoarchitectural visualization of in vivo and ex vivo animal and ex vivo human brain tumors.

Fluorescent contrast agents remain pivotal to CLE’s functionality. Advances in agent formulation and administration protocols have enhanced image clarity and diagnostic accuracy. These agents enhance visualization by selectively accumulating in tumor tissues, exploiting the disrupted blood–brain barrier commonly found in neoplasms. FNa, with its high affinity for tumor regions, produces bright fluorescence, enabling the identification of tumor margins and aiding in distinguishing healthy from pathological tissue (Figure 2) [53]. Similarly, ICG enhances vascular imaging, providing detailed insights into the tumor’s microvascular environment. Its near-infrared fluorescence allows for deeper tissue penetration and the visualization of subcellular structures [54]. The combination of these agents with advanced CLE technologies significantly improves diagnostic accuracy and intraoperative decision making. Their role in enhancing image clarity and real-time histopathological assessment underscores their importance in modern neurosurgical workflows, paving the way for more precise and effective brain tumor surgeries. Integration with other imaging technologies has also enhanced CLE’s utility. For instance, combining CLE with wide-field fluorescence imaging provides both macroscopic and microscopic insights, facilitating comprehensive tumor assessment [55].

## 5. Current Roles of CLE Visualization for Ex Vivo and In Vivo Human Brain Samples

Ex vivo brain tissue CLE-based visualization offers a practical and reliable approach for the detailed examination of excised brain tissue. This modality is particularly valuable in fluorescence-guided surgeries, in which it complements traditional histopathology by providing near-instantaneous imaging of tissue microstructures [56]. Belykh et al. [57] demonstrated the high specificity (94%) and positive predictive value (97%) of ex vivo CLE in identifying gliomas, underscoring its utility in distinguishing lesional tissue from healthy brain matter. By enabling an immediate analysis of excised tissue, ex vivo CLE reduces the risk of sampling errors inherent in conventional frozen sections. Furthermore, in cases of gliomas, ex vivo CLE has shown significant promise in grading tumors. Radtke et al. [58] highlighted its ability to identify critical histological markers such as necrosis, nuclear pleomorphism, and microvascular proliferation with an accuracy exceeding 90%. This capability is crucial for differentiating HGGs from their lower-grade counterparts, thereby guiding subsequent treatment strategies.

Building upon these findings, Byun et al. [50] demonstrated that the cCeLL-Ex vivo CLE system achieved a diagnostic accuracy of 89.2%, marginally surpassing that of frozen section analysis (86.5%). Moreover, it exhibited a superior specificity (70% compared to 50%) and significantly reduced the average diagnostic time to 13 min and 17 s, compared to 28 min and 28 s for frozen sections. These results underscore the potential of this system as a rapid, accurate, and reliable intraoperative diagnostic tool, complementing traditional cytological techniques while addressing the logistical and temporal constraints associated with frozen section analysis in neurosurgical practice.

However, ex vivo CLE is not without limitations. The process of excising and preparing tissue introduces delays that can disrupt surgical workflows. Moreover, image quality may degrade over time due to tissue handling and environmental factors.

On the other hand, intraoperative in vivo CLE addresses the limitations of ex vivo imaging by offering the immediate, real-time visualization of brain tissue. This capability significantly enhances surgical precision, allowing surgeons to differentiate between tumors and normal tissues during resection. Abramov et al. [59] demonstrated the feasibility of in vivo CLE in a clinical-grade system. This study demonstrated that high-resolution, real-time imaging could be achieved intraoperatively—with the first interpretable image obtained within about 5 s and after roughly six images—thus enabling rapid digital “optical biopsies”. Furthermore, the CLE images were directly correlated with both frozen and permanent histological sections, yielding diagnostic accuracies of 94% (versus frozen sections) and 92% (versus permanent histology), along with excellent sensitivity and specificity metrics. Importantly, the study employed neuronavigation to precisely document the biopsy location, ensuring that the CLE findings accurately corresponded to the histopathological samples. These methodological strengths underline the potential of in vivo CLE to streamline intraoperative decision making and possibly reduce reliance on traditional frozen section analyses. The ability to acquire interpretable images within seconds streamlines surgical workflows, reducing the reliance on frozen sections, which are not always available, especially in emergency settings. Xu et al. [53] further highlighted the advantages of in vivo CLE, particularly its superior image quality and diagnostic sensitivity compared to ex vivo imaging (Table 3).

However, despite its advantages, in vivo CLE is not without challenges. Motion artifacts, background fluorescence, and the grayscale nature of images can hinder interpretation. To address these issues, innovative approaches such as neural-network-based image style transfer have been employed. Izadyyazdanabadi et al. [70] demonstrated how style transfer techniques could transform CLE images to mimic the appearance of H&E staining, enhancing their interpretability for neuropathologists.

The integration of ex vivo and in vivo CLE into neurosurgical practice highlights their complementary strengths. Ex vivo CLE excels in post-resection analysis, providing detailed histological insights that guide diagnosis and treatment planning, without the risk of motion artifacts. In contrast, in vivo CLE serves as a real-time decision-making tool, offering immediate feedback on tumor margins and tissue characteristics, as demonstrated by Wagner et al. [39]. By comparing CLE with frozen sections, the authors demonstrated CLE had a comparable diagnostic accuracy and significantly reduced processing time (3 min for CLE vs. 27 min for frozen sections).

## 6. CLE for the Identification of Glioma Margins

Identifying glioma margins poses a significant challenge in neurosurgical oncology, in which the precise differentiation between tumor and normal tissue is crucial for optimizing patient outcomes [13]. The high image definition and magnification of CLE have emerged as promising features, offering cellular-level imaging during surgery. Recent advancements have demonstrated the value of CLE in intraoperative glioma management. Studies highlight CLE’s capacity to delineate infiltrative glioma margins with a diagnostic accuracy that approaches that of conventional histology. Abramov et al. [69] demonstrated its ability to identify regions of infiltration with a greater sensitivity compared to traditional fluorescence-guided imaging. This capability enables neurosurgeons to make informed decisions about resection boundaries during surgery, potentially improving the EOR while preserving critical structures.

CLE’s adaptability to intraoperative use lies in its ability to deliver rapid, high-resolution imaging in vivo. Visualizing cellular architecture and pathological changes provides neurosurgeons with actionable insights directly within the surgical field [49]. Xu et al. [40,67] emphasized CLE’s role in augmenting surgical precision by distinguishing glioma infiltration from adjacent healthy brain tissue. Despite these advancements, challenges such as motion artifacts and variability in image interpretation persist, necessitating the further optimization of CLE protocols and training.

Lastly, autofluorescence-based CLE shows promise for intraoperative glioma margin recognition, enabling label-free imaging of tissue microstructures. In recent years, the use of autofluorescence imaging has emerged as a promising, label-free modality for intraoperative tumor characterization. Unlike traditional fluorescence techniques that rely on exogenous dyes (e.g., 5-ALA or FNa), autofluorescence imaging capitalizes on the intrinsic fluorescent properties of brain tissue. These properties arise from endogenous fluorophores associated with cellular metabolism and structural components, such as NADH, flavins, and components of the extracellular matrix [71]. Notably, Reichenbach et al. demonstrated that label-free CLE could effectively capture distinctive autofluorescence patterns that differentiate tumor tissue from normal brain parenchyma [68]. Their study revealed that differences in spectral properties—such as peak positions and overall intensity—as well as spatial distribution patterns of autofluorescence, can serve as reliable indicators of tumor histopathology. These autofluorescence signatures enable the identification of key features, including hypercellularity, nuclear pleomorphism, and necrosis, which are critical for differentiating HGGs and LGGs.

In a complementary study, Radtke et al. showed that autofluorescence-based CLE not only facilitates real-time margin detection but also allows for the assessment of histomorphological characteristics without the need for conventional staining. This approach provides immediate diagnostic feedback by correlating specific autofluorescence signals with the underlying tissue architecture, thereby enhancing intraoperative decision making [58]. 

Furthermore, Picart et al. highlighted the potential of combining autofluorescence with induced fluorescence techniques. This integrated strategy leverages the strengths of both modalities—using the metabolic insights provided by autofluorescence alongside the high-contrast imaging of induced fluorescence—to improve the sensitivity and specificity of tumor detection and classification [41]. 

Together, these advances underscore the utility of autofluorescence imaging as a rapid, non-invasive tool for tumor-type identification.

## 7. Integration of CLE with Deep Learning

Interpreting in vivo CLE images presents challenges for untrained users due to motion artifacts, fluorescence signals exceeding the detector’s dynamic range, and obstructions caused by red blood cells. These factors often result in a substantial proportion of images being classified as non-diagnostic (ND). Despite these challenges, a single CLE image with a pathognomonic histological signature can provide a definitive intraoperative diagnosis. The vast amount of data generated by CLE—potentially hundreds to thousands of images per case—necessitates the development of efficient methods for data management and analysis. Deep learning, particularly through deep convolutional neural networks (DCNNs), has emerged as a transformative solution to these challenges [72,73]. 

DCNNs address several critical applications in CLE imaging, including diagnostic image detection, tumor classification, and feature localization [74]. These methods significantly enhance the diagnostic utility of CLE images, enabling more precise and tailored surgical interventions [72,75]. In diagnostic image detection, DCNNs can automatically classify CLE images as diagnostic or non-diagnostic, reducing the time and effort required for manual review. This capability is particularly valuable because many CLE images are deemed non-diagnostic due to artifacts or suboptimal quality [62]. Initial efforts to address this issue employed entropy-based approaches to filter non-diagnostic images. However, these methods were inadequate, as many ND images exhibit entropy levels similar to diagnostic ones [72,76]. In contrast, DCNN models have been developed to reliably classify images into diagnostic and ND categories using a training dataset based on evaluations by neuropathologists and neurosurgeons. These models achieve a high level of concordance with expert evaluations. For example, a study developing 42 new models, including 30 single and 12 ensemble models, demonstrated that DCNN-based evaluations achieved greater agreement with ground-truth annotations than entropy-based methods [72]. 

In tumor classification, DCNNs differentiate tissue types, such as tumor, injured brain, and normal brain tissue. This capability is particularly relevant as FNa may accumulate in injured areas due to blood–brain barrier disruption, potentially causing diagnostic ambiguity during surgery. A DCNN model inspired by the Inception architecture classified CLE images into these categories with a high accuracy, rivaling the diagnostic performance of trained neuropathologists. The model’s accuracy improved further when differentiating between tumor and non-tumor tissues, a distinction critical for surgical decision making [44,72,77]. Additionally, a cascaded deep decision network (DDN) achieved an 86% accuracy in classifying GBM images, outperforming traditional methods such as support vector machines (SVMs) and shallow CNNs [78,79]. 

Beyond classification, deep learning enhances CLE image quality by localizing diagnostically significant features. Weakly supervised learning (WSL) methods have been employed to highlight regions with diagnostic significance, such as areas of hypercellularity or abnormal nuclear features. For example, neuron activation maps generated by the first layer of a DCNN model emphasized regions with optimal image contrast, revealing the diagnostic tissue architecture. Similarly, a sliding window approach effectively identified clusters of malignant cells and hypercellular areas, offering transparent and objective diagnostic evidence [72,80]. 

Another approach utilized a global average pooling layer to segment images into diagnostic and ND regions, with results closely aligning with expert annotations [81,82]. 

In conclusion, deep learning techniques, particularly DCNNs, are crucial for maximizing the potential of CLE imaging. By automating the analysis of extensive datasets, DCNNs enhance the speed and accuracy of intraoperative diagnoses, addressing critical challenges such as non-diagnostic images, tumor classification, and feature localization. These advancements improve image interpretation, enabling more precise surgical interventions. Moreover, by integrating diagnostics with personalized therapeutic approaches, DCNNs contribute significantly to theranostics, ensuring better outcomes for patients with invasive malignant brain tumors.

## 8. Real-Time Telepathology as an Adjunct in the Surgical Theater

Telepathology employs digital systems to enhance communication between surgeons and pathologists, enabling remote diagnoses beyond the hospital environment [83]. However, these interactions have traditionally centered around and heavily relied upon frozen sections. To overcome these limitations and expand the scope of telepathology, recent advancements have introduced innovative technologies, including CLE [72]. Specifically, when integrated with CLE, telepathology represents a significant development in neurosurgery, improving intraoperative communication and decision making while addressing limitations of traditional diagnostic methods, such as discrepancies caused by imprecise or inadequate tissue sampling, artifacts generated during frozen section preparation, and errors arising from tumor heterogeneity across regions [84,85]. 

A key example of this innovation is the integration of CLE with a telepathology software platform (TSP) (Version 2024), which facilitates the remote visualization of the tissue microarchitecture without requiring a physical biopsy, enabling a seamless collaboration between pathologists and neurosurgeons across different locations [66]. Abramov et al. [66] illustrate the practicality of this approach in a study involving CLE and TSP during brain tumor surgeries. By enabling the transmission of CLE video-rate images in real time, the system supports immediate consultations, significantly shortening diagnostic timelines compared to frozen sections, which often require 20–45 min or more [86]. 

The CLE system is capable of generating usable histopathological images within 6 s, with an average usage time of 1 min per case. This rapid feedback provides a critical advantage, allowing neurosurgeons to make informed decisions promptly during procedures involving delicate and complex brain structures [66]. Moreover, the diagnostic accuracy of CLE, particularly with video-rate images, has been shown to reach 96%, notably surpassing the 63% accuracy of still CLE images. Dynamic visualization through video-rate not only enhances accuracy but also aids pathologists in distinguishing tumor cells from artifacts.

From an operational standpoint, the integration of CLE and TSP significantly enhances both efficiency and flexibility. Abramov et al. [66] describe how pathologists, either in loco or remotely, can interpret CLE images using a secure, cloud-based platform. The TSP ensures encrypted data transmission, safeguarding patient information while maintaining robust remote connectivity. This setup not only overcomes geographical barriers but also maximizes the utilization of specialist expertise, addressing the shortage of neuropathologists in various regions.

Maragkou et al. [87] further highlight the transformative potential of CLE in telepathology by emphasizing its compatibility with digital pathology advancements. Unlike traditional methods, CLE, as an inherently digital modality, eliminates delays associated with slide preparation and scanning. The instantaneous availability of digital images for teleconsultation streamlines intraoperative workflows and reduces reliance on physical biopsies, thereby minimizing procedural invasiveness. However, the clinical utility of CLE varies depending on specific indications. For instance, while frozen sections and cytology remain superior for evaluating sample adequacy and conducting ancillary diagnostic techniques, CLE offers distinct advantages for rapid preliminary insights. A hybrid approach combining CLE with frozen sections or cytology could optimize intraoperative diagnostic workflows [87]. 

The adoption of CLE telepathology also presents opportunities and challenges for neuropathologists. On one hand, it fosters closer collaborations with neurosurgeons, enhancing their role within the clinical team. On the other hand, it demands expertise in interpreting CLE images and adapting to real-time consultations. Comprehensive training in CLE-specific diagnostic criteria and techniques for managing motion artifacts or image quality issues is essential to fully harness this technology [66]. 

## 9. Limitations

Despite the promising benefits of the use of ex vivo and in vivo CLE in neuro-oncology, this new tool still presents limitations. It is important to acknowledge that current brain tumor classification increasingly relies on molecular alterations rather than solely on histological features. Although CLE has demonstrated remarkable utility in providing rapid, real-time imaging of cellular architecture and guiding intraoperative decision making, its capabilities are inherently limited to morphological assessment. Consequently, CLE may not detect high-grade lesions that are defined primarily by molecular features in the absence of overt histological abnormalities. For example, tumors that exhibit aggressive behavior based on genetic or epigenetic markers may appear deceptively indolent under CLE, leading to the potential underestimation of the tumor grade. Thus, while CLE can offer valuable insights into tissue architecture and delineate tumor margins, it should be considered an adjunct to conventional diagnostic modalities rather than a standalone tool.

Furthermore, although CLE-assisted tumor resection—utilizing the in vivo identification of glioma margins—has shown promising results, no prospective studies have yet been reported. Future investigations should therefore focus on evaluating the real-life impact of this intraoperative tool on overall survival and progression-free survival in these patient cohorts.

Therefore, CLE is best positioned as a complementary tool, augmenting frozen sections rather than replacing them entirely. Its capacity for rapid repetitive analyses is particularly valuable for evaluating tumor margins and guiding surgical resections [66]. 

## 10. Real-Life Benefits and Future Directions of CLE in Neuro-Oncology

CLE has emerged as a promising intraoperative tool, offering high-resolution, real-time imaging aiding surgeons in distinguishing tumor tissue from healthy brain tissue. CLE may assist neurosurgeons by providing real-time, high-resolution imaging of brain tissue at the cellular level. This advanced imaging technology allows surgeons to perform rapid “optical biopsies”, thereby enabling the immediate identification of tumor margins and distinguishing between neoplastic and healthy tissue. The ability to obtain the first interpretable images within seconds enhances intraoperative decision making and reduces the reliance on traditional frozen section analysis, which is often time-consuming and subject to sampling errors. By facilitating more precise tumor delineation, CLE supports maximal safe resection while minimizing the risk of damaging critical brain structures, ultimately improving patient outcomes. In real-life clinical practice, the high diagnostic accuracy of CLE—demonstrated in studies whose accuracies reached over 90%—translates into more confident surgical interventions. Additionally, the integration of CLE with telepathology platforms provides remote expert consultations, ensuring that even less experienced surgeons benefit from immediate, expert image interpretation. Recent enhancements, including video-rate imaging and the application of deep learning algorithms, further streamline the surgical workflow by reducing interpretative challenges and minimizing motion artifacts. Together, these advancements not only shorten operative times but also contribute to reduced postoperative complications and a faster recovery. Moreover, CLE use may reduce the frequency of reoperations by providing more accurate real-time feedback that enables clear resection margins and contributes to improved long-term survival, thereby enhancing overall patient care.

Additionally, CLE may represent a valuable tool in the in vivo identification of prognostic tissutal elements in neuro-oncology. Interestingly, extensive research has documented the role of tumor-associated microglia and macrophages (TAMs) in enhancing the malignancy of brain tumors [11]. Despite this, most clinical studies utilizing CLE have concentrated on the overall tissue architecture rather than on isolating specific cellular components. Recent preclinical studies have explored the potential of using targeted fluorescent probes—such as antibodies against microglial/macrophage markers (e.g., Iba1 and CD68)—to label and visualize these cells in situ with CLE [88,89]. These investigations indicate that it is technically feasible to identify TAMs during surgery; however, the approaches remain largely experimental and have yet to be widely validated or adopted in routine clinical practice.

Finally, in the current landscape of intraoperative diagnostics, several emerging techniques are complementing the established role of CLE in neurosurgical oncology. Ultra-fast deep learning CNS tumor classification, as recently described by Vermeulen et al., employs advanced neural network algorithms to deliver near real-time diagnostic information, thereby supporting the cellular-level insights provided by CLE [90]. Similarly, a novel method using label-free stimulated Raman scattering microscopy combined with a residual convolutional neural network has demonstrated rapid qualitative tumor detection without the need for exogenous dyes [91]. In addition, third-harmonic-generation microscopy integrated with deep learning offers fast histology-based diagnoses of gliomas, providing detailed structural data that further guide intraoperative decision making [92]. Moreover, intraoperative flow cytometry contributes quantitative assessments of tumor grade and extent of resection, adding another layer of diagnostic precision [93]. Although each of these techniques operates on distinct principles, their integration with CLE could establish a multimodal framework that harnesses both morphological and molecular insights. This combined approach promises to enhance the speed and accuracy of intraoperative tumor characterization, optimize surgical resection strategies, and ultimately improve patient outcomes.

## 11. Conclusions

CLE enhances intraoperative visualization by distinguishing malignant from healthy tissue in real time. Integrated fluorescence contrast agents enhance surgical precision during resection and tissue sampling. Deep learning reduces interpretive challenges, filtering non-diagnostic images and highlighting critical pathologies. Telepathology platforms further augment CLE’s impact by enabling remote, real-time neurosurgeon–pathologist collaborations, thus expediting decisions. Although CLE does not replace established techniques like frozen sections, it complements them, accelerating workflows and improving diagnostic accuracy. Ongoing refinements and larger clinical trials are poised to reinforce CLE’s role in neuro-oncological surgery, ultimately optimizing patient care.

## Figures and Tables

**Figure 1 diagnostics-15-00499-f001:**
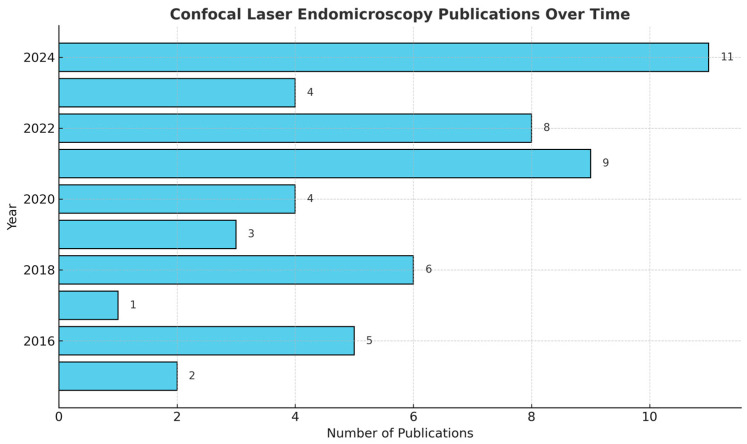
Time map of the publications investigating the roles and limitations of CLE in neuro-oncology over the years.

**Figure 2 diagnostics-15-00499-f002:**
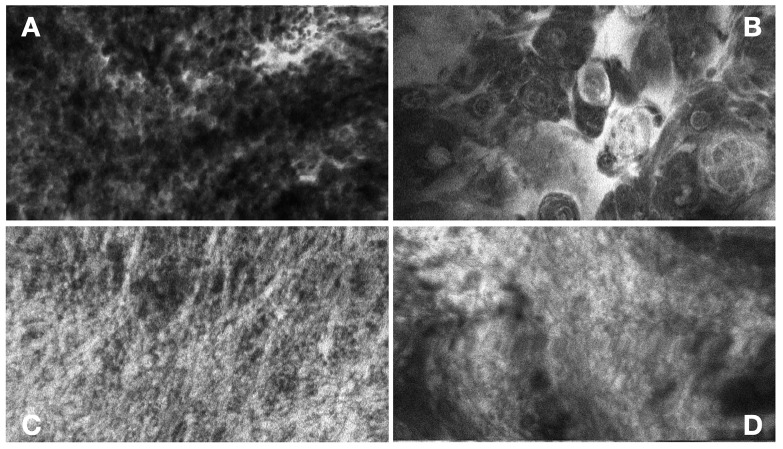
(**A**) Glioblastoma. High cellularity of pleomorphic tumor cells, with fluorescence enriched in tumor matrix. (**B**) Meningioma. Prominent psammoma bodies typical for psammomatous meningiomas. (**C**) Schwannoma. Pseudo-palisadal arrangement of tumor cells with parallel fibrous structures. (**D**) Metastasis of Squamous Cell Carcinoma. Squamous growth pattern of tumor cells, with fluorescence enhancement in tumor cells’ cytoplasms.

**Table 1 diagnostics-15-00499-t001:** Direct and indirect visualization tools in neuro-oncological surgery.

Direct Visualization Tools	Indirect Visualization Tools
Operating Microscope—Provides magnified, high-resolution real-time view of the surgical field.	iCT—Offers real-time cross-sectional imaging during surgery for guidance.
Neurosurgical Endoscope—Allows for direct visualization through narrow corridors.	iMRI—Provides real-time or near-real-time MRI to confirm resection extent or tissue changes.
Exoscope—High-definition digital “external scope” projecting a magnified surgical view onto a monitor.	Fluoroscopy/C-Arm—X-ray-based real-time imaging, often used for spinal instrumentation.
Intraoperative Ultrasound—Offers real-time imaging of brain structures with different densities.	Stereotactic Navigation Systems—Use pre- or intraoperative imaging (CT/MRI) to track instruments in 3D.
Intraoperative Angiography—Contrast-based imaging of vessels, often combined with fluoroscopic guidance.	
Surgical Loupes—Simple optical magnification worn by the surgeon for direct visualization.	Raman Spectroscopy—Analytical technology enabling the understanding the molecular composition of a tissue.
Near-Infrared Fluorescence Imaging—Assesses blood flow directly (e.g., ICG and FNa).	PET/CT—Enhances surgical management by accurately delineating metabolic activity, guiding the resection and biopsy of active tumor regions.
Intraoperative Optical Coherence Tomography (OCT)—Real-time “optical biopsy” for surface imaging.	
Confocal Laser Endomicroscopy—Enables microscopic “live” imaging of tissue at cellular resolution in real time.	

**Table 2 diagnostics-15-00499-t002:** Current fields of application of CLE in neuro-oncological surgery.

Field of Application	Roles
Tumor Margin Identification	Distinguishes between tumor and healthy brain tissue during surgery, aiding maximal resection while preserving critical structures [46].
Intraoperative Histological Analysis	Provides real-time optical biopsies, enabling rapid pathological assessment without traditional histology [46].
Glioma Surgery	Enhances precision in glioma resection by identifying infiltrative tumor margins, especially in eloquent areas [47].
Brain Metastases	Differentiates metastases from surrounding edema or reactive changes [48].
Guidance for Minimally Invasive Surgery	Integrated into endoscopic procedures for real-time tissue characterization during minimally invasive surgeries [49].
Surgical Training and Research	Provides a platform for neurosurgical education and exploring tissue pathology, advancing research on tumor biology [50].

**Table 3 diagnostics-15-00499-t003:** Summary of the main outcomes of the in vivo experiences of CLE.

Authors	Year	CLE Device	Type of Fluoroscope	Number of Patients	Pathologies	Main Outcome
Sanai et al. [60]	2011	Optiscan	5-ALA	10	10 LGGs	No macroscopically visible fluorescence, but CLE identified fluorescence at the cellular level in 100% of cases, aligning with tumor infiltration areas that were histopathologically confirmed.
Sanai et al. [37]	2011	Optiscan	FNa	33	13 LGGs 8 HGGs 8 Meningiomas 2 Metastasis 2 Other findings	Among HGGs, the microvascular proliferation was clearly identifiable and guided the resection, allowing the surgeon to distinguish the infiltrated margins from the healthy tissue. In LGGs, the alteration of cellular morphology overlapped with the pathological signal in T2W-MRI. Astrocytomas could be clearly distinguished from oligodendrogliomas. Concerning extra-axial tumors, CLE allowed the surgeon to differentiate among the different subgroups of meningiomas.
Eschbacher et al. [61]	2012	Optiscan	FNa	50	24 Meningiomas 12 HGGs 8 LGGs 4 Schwannomas 1 Ependymoma 1 Hemangioblastoma	CLE correlated well with histopathology, with 92.9% concordance in the blinded investigation.
Martirosyan et al. [62]	2016	Optiscan	FNa	74	30 Meningiomas 13 HGGs 8 LGGs 4 Schwannomas 2 Craniopharyngiomas 2 Hemangioblatomas 7 Other Tumors 8 Other findings	The use of CLE in glioma surgery demonstrated a 94% specificity and 91% sensitivity, while the traditional frozen section performed respectively at 95% and 94%.
Pavlov et al. [49]	2016	Cellvizio	5-ALA (3) FNa (6)	9	4 HGGs 4 LGGs 1 Lymphoma	CLE differentiated healthy and pathological tissue, without, however, allowing for an immediate histological classification. Fluorescein-based CLE offered clearer cellular patterns than 5-ALA.
Belykh et al. [63]	2018	Convivo	FNa	22	8 HGGs 3 Meningiomas 1 LGGs 2 Hemangioblastomas 1 Schwannoma 1 Craniopharyngioma 1 Metastasis 3 Other Findings 2 Other Tumors	CLE 3D imaging provided superior visualization of cellular architecture and vascular structures compared to 2D imaging, aiding in the identification of tumor margins. Informative image depths ranged from 9 to 28 µm in humans, supporting CLE’s utility for precise tumor resections.
Charalampaki et al. [64]	2019	Cellvizio	ICG	13	5 Gliomas 3 Meningiomas 2 Neurinomas 3 Metastasis	From a macroscopic point of view, ICG uptake was more superficial in meningiomas and metastases, limited in HGGs and absent in LGGs. Microscopically, CLE allowed for a clear identification of the margins and the detection of histological features (e.g., psammomatous bodies in meningiomas).
Hohne et al. [48]	2021	Convivo	FNa	12	5 Metastasis 4 HGGs 1 LGGs 2 Other Tumors	FNa uptake was observed in 100% of the operated tumors. CLE-specific features were highlighted in a practical clinical setting.
Xu et al. [53]	2022	Convivo	FNa	30	13 Gliomas 5 Meningiomas 3 Metastasis 1 Hemangioblastoma 1 Schwannoma 4 Other tumors 4 Treatment effect	In vivo CLE images exhibited statistically significantly higher levels of brightness and contrast; imaging quality decreased over time ex vivo. The in vivo images yielded a better quality in gliomas, meningiomas, and metastatic tumors.
Park et al. [65]	2022	Convivo	FNa	3	1 Metastasis 2 Gliomas	The feasibility of real-time visualization of CLE-images by a remotely located neuropathologist with real-time consultation with a neurosurgeon was investigated, and three explicative cases are described.
Abramov et al. [66]	2022	Convivo	FNa	11	6 Gliomas 1 Metastasis 3 Other Tumors 1 Other Finding	Video-rate CLE allowed for a correct interpretation of 96% of the biopsies, outperforming the still CLE images, which yielded an accuracy of 63% (*p* = 0.005).
Xu et al. [67]	2023	Convivo and Optiscan	FNa	28	26 HGGs 2 LGGs	Neurosurgeons and neuropathologists reviewed CLE images and assessed for possible margin infiltration using a numeric (1–5) and a dichotomous score. In the first case, neurosurgeons showed a higher level of overall agreement than neuropathologists (61% vs. 48%, *p* < 0.01). The difference in the diagnostic performance between neurosurgeons and neuropathologists was not statistically significant.
Reichenbach et al. [68]	2024	Convivo	No Contrast	3	1 Meningioma 1 GBM 1 Non tumor brain tissue	Autofluorescence patterns were analyzed. Spontaneous fluorescence was demonstrated in tumoral and non-tumoral tissue, with the diffuse fluorescence pattern being more present in pathological cells rather than normal cells, with only the exception of GBM, which often yielded a pattern with sparse, punctiform autofluorescence.
Abramov et al. [69]	2024	Convivo	FNa	33	26 HGGs 5 LGGs 2 Other Findings	CLE had higher sensitivity (0.73 vs. 0.38, *p* < 0.001), lower specificity (0.41 vs. 0.82, *p* = 0.03), and relatively similar PPV (0.79 vs. 0.86) and NPV (0.33 vs. 0.31) values compared to 5-ALA. For LGG, the sensitivity of CLE imaging was significantly higher (*p* = 0.006) but the level of specificity was significantly lower (*p* = 0.08) than that in 5-ALA imaging. For HGG, 5-ALA had a higher specificity (0.73 vs. 0.46, *p* = 0.37) but a lower sensitivity (0.43 vs. 0.70, *p* = 0.02) than CLE.
Wagner et al. [39]	2024	Convivo	FNa	203	49 Metastasis 9 LGGs 46 Meningiomas 84 HGGs 4 Schwannomas 2 Other Tumors 9 Other Findings	The diagnostic accuracy for CLE was inferior to frozen sections (0.87 vs. 0.91), failing to reach non-inferiority. The median time until intraoperative diagnosis was 3 min for CLE and 27 min for FS (*p* < 0.001). CLE achieved accuracies of 0.96 for LGGs, 0.93 for HGGs, 0.98 for meningiomas, 0.93 for metastases, and a mean accuracy of 0.95 across all tumors. In the meningioma subgroup analysis, CLE was not revealed to be inferior to frozen sections.

FNa: sodium fluorescein; HGG: high-grade glioma; LGG: low-grade glioma; ICG: indocyanine green; 5-ALA: 5-aminolevulinic acid.

## Data Availability

All data reported in this review are publicly available through the mentioned references.

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
