# Peer review of "Confocal Laser Endomicroscopy: Enhancing Intraoperative Decision Making in Neurosurgery"

_diagnostics, 2025, doi:10.3390/diagnostics15040499_

Round 1
Reviewer 1 Report
Comments and Suggestions for Authors
1. It would be appropriate to provide an optical diagram of an endomicroscope with characteristic images.
2. What are the estimates of patient life expectancy with intraoperative use of endomicroscopes?
3. It would be desirable to cover in more detail the autofluorescence approach to identifying the tumor type.
4. It would be appropriate to indicate not only the excitation wave length, but also the fluorescence for protoporphyrin IX.
5. How is it proposed to solve the problem of identifying deep-lying tissue areas for multiform glioblastomas?
6. There are publications on the active introduction of tumor-associated microglia and macrophages into tumor tissue, which increase the malignant characteristics of the tumor. Are there any results on their identification using an endomicroscope.
Author Response
- “It would be appropriate to provide an optical diagram of an endomicroscope with characteristic images.”
We thank the reviewer for this comment, and we completely agree. Therefore, we added Figure 2 to the manuscript, which we believe visually conveys CLE's real capabilities and resolution. Moreover, we added a brief description of the cases in the Figure caption.
- “What are the estimates of patient life expectancy with intraoperative use of endomicroscopes?”
We thank the reviewer for this comment as it allows us to clarify this concept in our manuscript. To date, no prospective experiences investigating the impact of CLE-assisted tumor resection on overall survival or progression-free survival has been reported. Therefore, we added the following paragraph in the new Limitations section:
Lines 598-602.
“Although CLE-assisted tumor resection — utilizing in vivo identification of glioma margins — has shown promising results, no prospective studies have yet been reported. Future investigations should therefore focus on evaluating the real-life impact of this intraoperative tool on overall survival and progression-free survival in these patient cohorts.”
- “It would be desirable to cover in more detail the autofluorescence approach to identifying the tumor type.”
We thank the reviewer for this comment and have enhanced the manuscript with additional information regarding the role of autofluorescence observed with CLE as an additional support for identifying tumor type.
Lines 314-340.
“In recent years, the use of autofluorescence imaging has emerged as a promising, label‐free modality for intraoperative tumor characterization. Unlike traditional fluorescence techniques that rely on exogenous dyes (e.g., 5‑ALA or FNa), autofluorescence imaging capitalizes on the intrinsic fluorescent properties of brain tissue. These properties arise from endogenous fluorophores associated with cellular metabolism and structural components, such as NADH, flavins, and components of the extracellular matrix. Notably, Reichenbach et al. demonstrated that label‐free CLE could effectively capture distinctive autofluorescence patterns that differentiate tumor tissue from normal brain parenchyma. Their study revealed that differences in spectral properties — such as peak positions and overall intensity — as well as spatial distribution patterns of autofluorescence, can serve as reliable indicators of tumor histopathology. These autofluorescence signatures enable the identification of key features, including hypercellularity, nuclear pleomorphism, and necrosis, which are critical for differentiating HGG and LGG.
In a complementary study, Radtke et al. showed that autofluorescence-based CLE not only facilitates real-time margin detection but also allows for the assessment of histomorphological characteristics without the need for conventional staining. This approach provides immediate diagnostic feedback by correlating specific autofluorescence signals with underlying tissue architecture, thereby enhancing intraoperative decision-making.
Furthermore, Picart et al. highlighted the potential of combining autofluorescence with induced fluorescence techniques. This integrated strategy leverages the strengths of both modalities — using the metabolic insights provided by autofluorescence alongside the high-contrast imaging of induced fluorescence — to improve the sensitivity and specificity of tumor detection and classification.
Together, these advances underscore the utility of autofluorescence imaging as a rapid, non-invasive tool for tumor-type identification.”
- “It would be appropriate to indicate not only the excitation wave length, but also the fluorescence for protoporphyrin IX.”
We thank the reviewer for this comment und have revised the manuscript accordingly.
Lines 107.
“(emission wavelength: 620 to 704 nm)”
- “How is it proposed to solve the problem of identifying deep-lying tissue areas for multiform glioblastomas?”
We thank the reviewer for this comment. We addressed this issue in section 4 “Advancement in CLE Technology”, where we discussed the Z-stack capabilities of the 2-gen CLE tools. We now further detailed the potential of Z-stack in the identification of deep-layered structures, especially in glioblastoma.
Lines 193-197.
“In addition, combining this approach with optimized fluorescent contrast agents and deep learning algorithms for real-time image interpretation further enhances the detection of infiltrative tumor margins. These integrated strategies may improve intraoperative decision-making and ensure a more complete and safe tumor resection.”
- “There are publications on the active introduction of tumor-associated microglia and macrophages into tumor tissue, which increase the malignant characteristics of the tumor. Are there any results on their identification using an endomicroscope”
We thank the reviewer for this very interesting comment. We are not aware of in vitro experiences with CLE for identifying tumor-associated microglia injected into tumor tissue, however we believe this would be an extremely interesting investigation topic. We revised the manuscript by adding a new section: Future directions.
Lines 631-641.
“CLE has emerged as a promising intraoperative tool, offering high‐resolution, real‑time imaging aiding surgeons in distinguishing tumor tissue from healthy brain tissue. […]
Additionally, CLE may represent a valuable tool in the in vivo identification of prognostic tissutal elements in neuro-oncology. Interestingly, extensive research has documented the role of tumor-associated microglia and macrophages (TAMs) in enhancing the malignancy of brain tumors. Despite this, most clinical studies utilizing CLE have concentrated on the overall tissue architecture rather than on isolating specific cellular components. Recent preclinical studies have explored the potential of using targeted fluorescent probes — such as antibodies against microglial/macrophage markers (e.g., Iba1 and CD68) — to label and visualize these cells in situ with CLE. These investigations indicate that it is technically feasible to identify TAMs during surgery; however, the approaches remain largely experimental and have yet to be widely validated or adopted in routine clinical practice.”
Reviewer 2 Report
Comments and Suggestions for Authors
Review Confocal Laser Endomicroscopy: Enhancing Intraoperative 2 Decision-Making in Neurosurgery
Well-written and informative review on the use of Confocal Laser Endomicroscopy in neurosurgery.
A figure with a few CLE representative images would be a good addition.
Table 1: The distinction between direct and indirect visualization tools is not clear, sounds arbitrary to place OCT and US in different categories as well as Near-Infrared Fluorescence Imaging and Angiography, for example. Please explain.
l.134: In neurosurgery, CLE has proven to be a transformative tool…. With a reference to figure 1 showing the number of publications increasing to 11 in 2024. If this should evidence ‘transformative’, it is not very strong, overstated.
The authors use video-flow several times, probably they mean video-rate.
The Abramov study, ref 59, could use some extra discussion. Until line 220 ex-vivo CLE studies comparing with frozen section are discussed, on human or murine samples, whereas ref 59 performed an in-vivo study where images were compared with corresponding frozen and permanent histology sections, with image correlation to biopsy location using neuronavigation.
Also, this should be followed up with a discussion on the clinical benefit of use of CLE, which clinical studies have been initiated?
Next I miss
- a discussion on the limitation that currently brain tumors types are defined by molecular alterations rather than by histology or by any other morphological technique including CLE. CLE may face similar limitations regarding lesions that are considered high-grade solely due to molecular features.
- Short para on CLE in the context of other emerging methods, e.g. Ultra-fast deep-learned CNS tumour classification during surgery, Nature 2023; Novel rapid intraoperative qualitative tumor detection by a residual convolutional neural network using label-free stimulated Raman scattering microscopy. Acta Neuropathol Commun. 2022; Fast intraoperative histology-based diagnosis of gliomas with third harmonic generation microscopy and deep learning, Scientific Reports 2022; Assessment of Gliomas' Grade of Malignancy and Extent of Resection Using Intraoperative Flow Cytometry, Cancers 2023, to name a few.
Author Response
- “A figure with a few CLE representative images would be a good addition.”
We thank the reviewer for this comment, and we completely agree. Therefore, we added Figure 2 to the manuscript, which we believe visually conveys CLE's real capabilities and resolution. Moreover, we added a brief description of the cases in the Figure caption.
- “Table 1: The distinction between direct and indirect visualization tools is not clear, sounds arbitrary to place OCT and US in different categories as well as Near-Infrared Fluorescence Imaging and Angiography, for example. Please explain.”
We thank the reviewer for this comment as it allows us to improve the quality of Table 1. As suggested, we revised the table to more accurately reflect the distinction of these tools used in everyday practice. We agree that US and OCT should be in the same category and revise the position of intraoperative angiography as should be rightfully acknowledged in the direct visualization tools. Furthermore, we added PET-CT scans to the Table, as this provides valuable information regarding the metabolic activity of the lesion, which improves the precision of biopsy and resection.
- “134: In neurosurgery, CLE has proven to be a transformative tool…. With a reference to figure 1 showing the number of publications increasing to 11 in 2024. If this should evidence ‘transformative’, it is not very strong, overstated.”
We thank the reviewer for this comment and agree. Therefore, we revised the manuscript to maintain a more cautious tone.
- “The authors use video-flow several times, probably they mean video-rate.”
We thank the reviewer for having identified this error in the manuscript and have revised it accordingly.
- “The Abramov study, ref 59, could use some extra discussion. Until line 220 ex-vivo CLE studies comparing with frozen section are discussed, on human or murine samples, whereas ref 59 performed an in-vivo study where images were compared with corresponding frozen and permanent histology sections, with image correlation to biopsy location using neuronavigation.”
We thank the reviewer for this comment and agree that the study by Abramov et al. should be further discussed. Therefore we added the following paragraph in the section Current Roles of CLE Visualization.
Lines 257-267.
This study demonstrated that high‐resolution, real-time imaging could be achieved intraoperatively — with the first interpretable image obtained within about 5 seconds and after roughly 6 images — thus enabling rapid digital “optical biopsies.” Furthermore, the CLE images were directly correlated with both frozen and permanent histological sections, yielding diagnostic accuracies of 94% (versus frozen sections) and 92% (versus permanent histology), along with excellent sensitivity and specificity metrics. Importantly, the study employed neuronavigation to precisely document the biopsy location, ensuring that the CLE findings accurately corresponded to the histopathological samples. These methodological strengths underline the potential of in vivo CLE to streamline intraoperative decision-making and possibly reduce reliance on traditional frozen section analyses.
- “Also, this should be followed up with a discussion on the clinical benefit of use of CLE, which clinical studies have been initiated?”
We thank the reviewer for this comment and agree that a focused section should summarize the real-world benefits of CLE. Therefore we added this information in the new section Real Life Benefits and Future Directions of CLE in Neuro-oncology.
Lines 609-630.
CLE may assist neurosurgery by providing real-time, high-resolution imaging of brain tissue at the cellular level. This advanced imaging technology allows surgeons to perform rapid “optical biopsies,” thereby enabling immediate identification of tumor margins and distinguishing between neoplastic and healthy tissue. The ability to obtain the first interpretable images within seconds enhances intraoperative decision-making and reduces the reliance on traditional frozen section analysis, which is often time-consuming and subject to sampling errors. By facilitating more precise tumor delineation, CLE supports maximal safe resection while minimizing the risk of damaging critical brain structures, ultimately improving patient outcomes. In real-life clinical practice, the high diagnostic accuracy of CLE—demonstrated in studies where accuracies reached over 90%—translates into more confident surgical interventions. Additionally, the integration of CLE with telepathology platforms provides remote expert consultation, ensuring that even less experienced surgeons benefit from immediate, expert image interpretation. Recent enhancements, including video-rate imaging and the application of deep learning algorithms, further streamline the surgical workflow by reducing interpretative challenges and minimizing motion artifacts. Together, these advancements not only shorten operative times but also contribute to reduced postoperative complications and faster recovery. Moreover, CLE use may reduce the frequency of reoperations by providing more accurate real-time feedback that enables clear resection margins and contributes to improved long-term survival, thereby enhancing overall patient care.
- “Next I miss a discussion on the limitation that currently brain tumors types are defined by molecular alterations rather than by histology or by any other morphological technique including CLE. CLE may face similar limitations regarding lesions that are considered high-grade solely due to molecular features.”
We thank the reviewer for this comment and revised the manuscript to acknowledge these limitations of CLE in neuro-oncology by adding a new section on Limitations.
Lines 523-533.
It is important to acknowledge that current brain tumor classification increasingly relies on molecular alterations rather than solely on histological features. Although CLE has demonstrated remarkable utility in providing rapid, real-time imaging of cellular architecture and guiding intraoperative decision-making, its capabilities are inherently limited to morphological assessment. Consequently, CLE may not detect high-grade lesions that are defined primarily by molecular features in the absence of overt histological abnormalities. For example, tumors that exhibit aggressive behavior based on genetic or epigenetic markers may appear deceptively indolent under CLE, leading to potential underestimation of tumor grade. Thus, while CLE can offer valuable insights into tissue architecture and delineate tumor margins, it should be considered an adjunct to conventional diagnostic modalities rather than a standalone tool.
- “Short para on CLE in the context of other emerging methods, e.g. Ultra-fast deep-learned CNS tumour classification during surgery, Nature 2023; Novel rapid intraoperative qualitative tumor detection by a residual convolutional neural network using label-free stimulated Raman scattering microscopy. Acta Neuropathol Commun. 2022; Fast intraoperative histology-based diagnosis of gliomas with third harmonic generation microscopy and deep learning, Scientific Reports 2022; Assessment of Gliomas' Grade of Malignancy and Extent of Resection Using Intraoperative Flow Cytometry, Cancers 2023, to name a few.”
We thank the reviewer for this comment as it allows us to enhance the quality of the manuscript by providing the readers with a framework of the current research in this field.
Lines 643-662.
In the current landscape of intraoperative diagnostics, several emerging techniques are complementing the established role of CLE in neurosurgical oncology. Ultra-fast deep-learned CNS tumor classification, as recently described by Vermeulen et al., employs advanced neural network algorithms to deliver near real-time diagnostic information, thereby supporting the cellular-level insights provided by CLE. Similarly, a novel method using label-free stimulated Raman scattering microscopy combined with a residual convolutional neural network has demonstrated rapid qualitative tumor detection without the need for exogenous dyes. In addition, third harmonic generation microscopy integrated with deep learning offers fast histology-based diagnosis of gliomas, providing detailed structural data that further refines intraoperative decision-making. Moreover, intraoperative flow cytometry contributes quantitative assessments of tumor grade and extent of resection, adding another layer of diagnostic precision. Although each of these techniques operates on distinct principles, their integration with CLE could establish a multimodal framework that harnesses both morphological and molecular insights. This combined approach promises to enhance the speed and accuracy of intraoperative tumor characterization, optimize surgical resection strategies, and ultimately improve patient outcomes.